# l-Proline Alleviates Kidney Injury Caused by AFB1 and AFM1 through Regulating Excessive Apoptosis of Kidney Cells

**DOI:** 10.3390/toxins11040226

**Published:** 2019-04-16

**Authors:** Huiying Li, Songli Li, Huaigu Yang, Yizhen Wang, Jiaqi Wang, Nan Zheng

**Affiliations:** 1State Key Laboratory of Animal Nutrition, Institute of Animal Science, Chinese Academy of Agricultural Sciences, Beijing 100193, China; thufit2012@126.com (H.L.); lisongli@caas.cn (S.L.); yanghgxms@163.com (H.Y.); m13613613572_2@163.com (Y.W.); wangjiaqi@caas.cn (J.W.); 2Key Laboratory of Quality & Safety Control for Milk and Dairy Products of Ministry of Agriculture and Rural Affairs, Institute of Animal Science, Chinese Academy of Agricultural Sciences, Beijing 100193, China; 3Laboratory of Quality and Safety Risk Assessment for Dairy Products of Ministry of Agriculture and Rural Affairs, Institute of Animal Sciences, Chinese Academy of Agricultural Sciences, Beijing 100193, China

**Keywords:** aflatoxin B1, aflatoxin M1, l-proline, proline dehydrogenase, apoptosis

## Abstract

The toxicity and related mechanisms of aflatoxin B1 (AFB1) and aflatoxin M1 (AFM1) in the mouse kidney were studied, and the role of l-proline in alleviating kidney damage was investigated. In a 28-day toxicity mouse model, thirty mice were divided into six groups: control (without treatment), l-proline group (10 g/kg body weight (b.w.)), AFB1 group (0.5 mg/kg b.w.), AFM1 (3.5 mg/kg b.w.), AFB1 + l-proline group and AFM1 + l-proline group. Kidney index and biochemical indicators were detected, and pathological staining was observed. Using a human embryonic kidney 293 (HEK 293) cell model, cell apoptosis rate and apoptotic proteins expressions were detected. The results showed that AFB1 and AFM1 activated pathways related with oxidative stress and caused kidney injury; l-proline significantly alleviated abnormal expressions of biochemical parameters and pathological kidney damage, as well as excessive cell apoptosis in the AF-treated models. Moreover, proline dehydrogenase (PRODH) was verified to regulate the levels of l-proline and downstream apoptotic factors (Bax, Bcl-2, and cleaved Caspase-3) compared with the control (*p* < 0.05). In conclusion, l-proline could protect mouse kidneys from AFB1 and AFM1 through alleviating oxidative damage and decreasing downstream apoptosis, which deserves further research and development.

## 1. Introduction

Aflatoxins B1 (AFB1) and aflatoxins M1 (AFM1) belong to a group of secondary metabolites with high toxicity synthesized by *Aspergillus flavus* and *Aspergillus parasiticus*, which can be produced in peanuts, corn, cottonseed, walnut, sesame, etc. [1,2]. AFB1 and AFM1 have similar chemical structures; AFM1 can be derived from AFB1 (4-hydroxy derivative of AFB1, Appendix A) in the liver by hepatic microsomal cytochrome P450, and can then enter the blood circulation and be excreted via lactation into milk [3]. The two aflatoxins are regarded as the most common and carcinogenic members in AF family, and the International Agency for Research on Cancer organization (IARC) clearly suggested that AFB1 and AFM1 should be classified to be a Group I carcinogens [4].

Several researchers have reported that they can lead to Reyes syndrome, neuropathy, respiratory diseases, renal injury, and liver damage [5,6]. The metabolic course of AFB1 and AFM1 are proved to be principally in the liver tissue [7,8,9]; lung and gastrointestinal tract are two other key targets which are susceptible to the carcinogenic effects of AFs [10,11,12]. AFB1 and AFM1 are also verified to be a pathogenic factor leading to underweight and hypoimmune young people [13,14]. In our recent study, AFB1 and AFM1 were found to be toxic to mouse kidneys, leading to abnormal expression of biochemical indicators related to renal function, as well as pathological staining of kidney tissue [15]. The mouse kidneys were analyzed with metabonomics technology, and l-proline was found to be significantly down-regulated. l-proline has been demonstrated to be down-regulated with the treatment of AFB1 and AFM1 (VENN plot, Appendix A). Upstream regulator proline dehydrogenase (PRODH) and downstream apoptosis factors were measured to investigate the toxicity mechanism of AFs [15]. However, the role of l-proline in protecting kidneys from the toxicity of AFB1 and AFM1 is still unclear, and will be explored in the present manuscript and expanded studies.

The purpose of the present study was to further investigate the protective effect of l-proline in alleviating the kidney damage caused by AFB1 and AFM1 in a mouse model, and to validate the role of the upstream regulator PRODH in regulating expression of l-proline and downstream apoptosis factors. Considering that food contamination is the main source of AFs, and AFB1 and AFM1 show strong toxicity to mammals, how to block the oral intake of AF-contaminated food and develop therapeutic drugs becomes necessary and urgent. Thus, with further safety and pharmacokinetics research, l-proline administration or its use in combination with other toxicides might be an alternative strategy for the clinical treatment of kidney injury caused by AFs.

## 2. Results

### 2.1. l-Proline Alleviates the Decreased Viability of HEK 293 cells Caused by AFB1 and AFM1

To evaluate the effects of AFB1 and AFM1 on kidney cells, as well as to investigate the role of l-proline in alleviating cell death caused by AFB1 and AFM1, HEK 293 cells were cultured, and the viability of the cells was detected by a cell counting (CCK8) kit. To exclude the effect of l-proline on HEK 293 cells, the cell viability in single l-proline group was detected, and results demonstrated that there was no obvious difference in cell viability between the l-proline (1 g/L, 97.4%) group and the control group (99.3%). At the same concentration (100 mg/L), AFB1 showed a stronger inhibitory effect (56.4% viability in the 100 mg/L group) than AFM1 (78.3% viability in the 100 mg/L group) (all *p* < 0.05 vs. control). With the treatment of l-proline (1 g/L), the viability in the AFB1 + l-proline or AFM1 + l-proline groups were 71.0% or 86.1%, respectively, which were significantly higher than the viability of AFB1 or AFM1 only groups (*p* < 0.05). The above results detected by CCK8 kit suggested that AFB1 and AFM1 pose toxic effects on HEK 293 cells and finally lead to cell death, and l-proline may protect the cells from the toxicity caused by AFB1 and AFM1 (Figure 1).

### 2.2. l-Proline Decreases Apoptosis Rate of HEK 293 Cells Induced by AFB1 and AFM1

In cell apoptosis detection by fluorescence-activated cell sorting (FACS), LL stands for alive cells, LR stands for early apoptosis cells, UR stands for late apoptosis cells and dead cells, respectively. Cell apoptosis detection by AnnexinV/PI assay showed that AFB1 and AFM1 significantly enhanced apoptosis rates when compared with the control group. With the addition of l-proline, cell apoptosis rates and death rates in the AFM1 + l-proline and AFB1 + l-proline groups were significantly down-regulated when compared with the AFM1 and AFB1 only groups (*p* < 0.05) (Figure 2).

### 2.3. l-Proline Alleviates the Kidney Damage Caused by AFB1 and AFM1, Referring to Both Biochemical Parameters and Pathology Condition

To evaluate the effects of the AFs and l-proline on kidney function, kidney tissue and serum were collected, several biochemical markers were measured by ELISA kits, and pathological kidney condition was observed. No matter in kidney tissue or in mice serum (Figure 3), these indicators showed no obvious difference between the control and the l-proline group, and the levels of creatinine (CRE), urea (UREA), uric acid (UA) in the AFB1 and AFM1 treatment groups were higher than the control (all *p* < 0.05 vs. control). The three indicators in the combination groups were significantly lower than the ones in AFB1 and AFM1 groups (*p* < 0.05), indicating that AFB1 and AFM1 led to kidney damage, which was in accordance with our previous detection results in mouse serum [15]. Additionally, l-proline was proved to alleviate renal injury (Figure 3). To investigate the effects of AFs and l-proline on oxidative stress in kidney, total antioxidant capacity (T-AOC) and malondialdehyde (MDA) were analyzed in kidney tissue, and results demonstrated that the level of T-AOC significantly decreased, and the level of MDA markedly increased (all *p* < 0.05 vs. control), indicating that l-proline might be helpful in alleviating oxidative damage and in regulating the levels of T-AOC and MDA compared to the AF groups (*p* < 0.05) (Figure 3).

To further investigate the effects of AFs and l-proline on the kidney, haematoxylin and eosin (H&E) stained histological sections were observed and analyzed, through scanning the immunohistochemistry (IHC) figures using the scoring system. From one mouse, 2 IHC figures were chosen from each IHC slide, and 10 IHC figures in total were chosen in each group (5 mice). There seemed no obvious damage of kidney tissue in the control group (scanning score: 0.46 ± 0.16) and l-proline group (scanning score: 0.43 ± 0.089). Compared with the control group, edema, cytomorphosis, severe inflammatory cell infiltration and large-area hemorrhage (Figure 4A) could be found in several areas of sections in the AFB1 group (scanning score: 3.47 ± 0.43) and the AFM1 group (scanning score: 3.22 ± 0.41), the scanning scores in AF groups were significantly higher than the ones in the control and l-proline groups (*p* < 0.05) (Figure 4B), indicating that single AF treatment significantly damaged kidney tissue. Compared to the AFs + l-proline combination groups, the damage degrees in AFB1 + l-proline group (scanning score: 2.28 ± 0.36) and AFM1 + l-proline group (scanning score: 2.20 ± 0.28) were significantly less severe (*p* < 0.05) (Figure 4B), and embodied on occasional edema, cytomorphosis and hemorrhage (Figure 4A). Results revealed that aflatoxins caused obvious damage in kidney tissue, and l-proline alleviated the degree of injury in kidney tissue.

### 2.4. Validation of the Protective Effect of l-Proline in Regulating Apoptosis Factors via PRODH

To investigate the toxic mechanism of the two AFs, as well as to further explore the protective mechanism of l-proline in alleviating kidney damage caused by AFs, the expression of PRODH and several apoptosis factors were measured both in kidney tissue and in HEK 293 cells. Western blot results demonstrated that treatment of AFB1/AFM1 significantly increased the levels of PRODH, Bax, and cleaved Caspase-3 (*p* < 0.05 vs. the control), while expression of Bcl-2 decreased (*p* < 0.05 vs. the control). With the addition of l-proline, the levels of PRODH, Bax, and cleaved Caspase-3 proteins were sharply down-regulated, and Bcl-2 was up-regulated when compared with the AFB1/AFM1 groups (*p* < 0.05) (Figure 5A). The results indicated that PRODH might be the target of the AFs, and l-proline could alleviate the excessive apoptosis caused by AFB1/AFM1.

After transfection with PRODH siRNA, the levels of Bax and cleaved Caspase-3 in the control/ AFB1/ AFM1 groups were markedly lower (*p* < 0.05) than in the normal cells, however, the levels of Bax and cleaved Caspase-3 in AFs + l-proline groups were significantly higher than the AF groups (*p* < 0.05) (Figure 5B). The results proved that PRODH might be a direct target of AFB1/AFM1, which was responsible for activating downstream apoptosis factors in normal kidney cell. Meanwhile, l-proline could alleviate kidney damage by inhibiting the excessive apoptosis of normal kidney cells.

## 3. Discussion

In several studies, AFB1 and AFM1 were proved to be excreted mainly through the biliary duct and the urinary pathway; AFB1 could also be detected in the kidney tissue and urine sample in calf models [16], which indicated the liver and the kidney were two main target organs of toxic AFs. There was direct evidence for the exposure of humans to AFB1/AFM1 by ingestion in several countries through identifying AFB1/AFM1 or their metabolites in human biological samples [17,18]. Because AFs are Group I carcinogens and widely exist in food contamination [4], their adverse effects and the related mechanism should be explored thoroughly in animal models. Considering that clinical research for specific detoxifying medicines for AFB1/AFM1 is rare, the research and development of curative medicines for AFs is becoming a necessary and urgent issue. In our recent research, AFB1 and AFM1 were found to pose toxic effects on mouse kidneys through activating PRODH and downstream apoptosis factors; l-proline was selected as the specific metabolite for AFB1/AFM1 in kidney tissue, which could significantly protect kidney damage from AFB1 and AFM1. 

The CCK-8 assay showed that l-proline, combined with the AFs, increased HEK 293 cell viabilities compared with aflatoxin in the absence of l-proline. The FACS assay showed that l-proline reduced AF related apoptosis and death of kidney cells. Together, these results suggested that AFB1 and AFM1 damaged the kidney by increasing cell apoptosis and death and that l-proline protected cell viability and alleviated excessive cell apoptosis, which was caused by AFB1/AFM1. 

Biochemical indicators including creatinine (CRE in kidney or Scr in serum), UREA, and UA were involved in inflammatory reactions, cell necrosis and toxicosis in several studies [19,20,21,22,23]; these were down-regulated with the addition of l-proline in the present research, and thus, we suggest that l-proline might play protective effects on kidney cells through alleviating inflammation, necrosis and toxicosis, which was validated by the following mechanism.

In addition, in AFs-treated groups, MDA in kidney tissue markedly increased, and T-AOC was significantly lower. l-proline was found to down-regulate MDA levels and up-regulate T-AOC levels compared to the AF-treated groups. MDA is usually produced from lipid peroxidation and is proved to be toxic to organ tissue [24], while T-AOC reflects the overall anti-oxidative activity of the organism [25] and is also relevant to lipid peroxidation in some degree [26]. The effects of AFB1 and AFM1 on these two indicators are consistent with their activation of oxidative reactions in mouse kidneys. Thus, l-proline might play anti-oxidation and anti-peroxidation roles in protecting the kidney tissue from oxidative damages.

The histological findings by H&E staining were consistent with kidney damage caused by AFB1 and AFM1, which were in accordance with the biochemical parameter data. Together, the above results proved that the kidney was one of the main toxicity targets of AFs, and indicated that some metabolites might be transferred, degraded or accumulated in kidney tissue. Proline was validated to be down-regulated with the treatment of AFB1 and AFM1 in a previous study and was proved to protect the kidney from oxidative injuries caused by AFB1 and AFM1 in the present study.

Considering the bioactivities of proline, several researchers reported that in addition to providing energy, proline has been found to alleviate oxidative stress in various models [27,28,29,30]. Krishnan et al. found that proline protected cells against tert-butyl hydroperoxide, H_2_O_2_ and carcinogenic oxidative stress inducers [31]. Rai et al. proved that proline alleviated damage from reactive oxygen species when cells were placed under metal stress [32]. Proline was also verified to improve oxidative stress tolerance in *E. coli* by a pre-adaptive effect, which might be related to the enhancement of catalase-peroxidase bioactivity and the production of endogenous hydrogen peroxide [33]. Therefore, after tests of the anti-oxidative activity of l-proline in animal models, it is meaningful to develop l-proline as a novel detoxifying drug for clinical use. 

Studies found that proline, in a flavoenzyme containing PRODH in a special polypeptide, transferred the oxidation of l-proline to glutamate in Gram-negative bacteria [34]. In our previous study, the expression level of l-proline, which was negatively regulated by PRODH, increased in response to AFs, along with apoptosis factors cleaved caspase-3 and Bax. In the present study, l-proline was verified to alleviate the excessive cell apoptosis caused by AFs, demonstrated by the lower levels of cleaved caspase-3 and Bax. Additionally, PRODH SiRNA treatment was performed to investigate whether PRODH is the direct target of AFB1 and AFM1. The results demonstrated that the pro-apoptotic factors were no different in AFs-treated and PRODH SiRNA-treated cells, however, in cells which were treated with PRODH SiRNA + AFs + l-proline group, these pro-apoptotic proteins seemed to be up-regulated when compared with AF-treated or PRODH SiRNA-treated groups. Combining the apoptosis detection results in both the FACS and western blot, we found that a single treatment of l-proline showed no apparent effect on cell apoptosis, while it regulated excessive apoptosis and protected the kidney tissue in AF-treated models, which further suggested that l-proline might have dual-directional regulations. 

In summary, we have identified l-proline as the key sensor in mouse kidneys tissue in AFs-induced toxicity model and proved that l-proline could protect kidney function through regulating the excessive apoptosis of kidney cells. We have also proved the activity of the upstream sensor PRODH, which regulates the level of l-proline, leading to kidney damage through the induction of oxidative stress and apoptosis. The findings will be helpful in the research and development of l-proline in preclinical and clinical fields.

## 4. Materials and Methods

### 4.1. Chemicals

Ninety-five percent pure AFB1 and AFM1 were purchased from Pribolab (Singapore). HEK 293 cells (a human epithelial kidney cell line) was obtained from the American Type Culture Collection Cells (ATCC, Manassas, VA, USA). Dulbecco’s Modified Eagle Medium (DMEM) and fetal bovine serum (FBS) were purchased from GIBCO (USA), l-glutamine was purchased from ChemCatch (USA), and 1% penicillin/streptomycin was purchased from Thermo Fisher (Waltham, MA, USA). l-proline with ninety-five percent purity was purchased from Sigma (St. Louis, MO, USA). Cell counting kit-8 (CCK-8 kit) was purchased from Dojindo (Kumamoto, Japan). Annexin V/PI staining kit for cell apoptosis detection was purchased from Solarbio (Beijing, China). SiRNA transfection kit was purchased from Solarbio. ELISA detection kits for CRE, UREA, UA, MDA and T-AOC in mouse kidney tissue and mouse serum were purchased from Jiancheng (Nanjing, China). A hematoxylin and eosin (H&E) staining kit, a total protein extraction kit, and PBST buffer (99.9% phosphate buffered solution + 0.1% Tween-20) were purchased from Solarbio. The primary antibodies of β-actin, PRODH, Bcl-2 Associated X Protein (Bax), cleaved cysteinyl aspartate specific proteinase 3 (cleaved Caspase-3), as well as the secondary antibodies, were purchased from Santa Cruz (Santa Cruz, CA, USA), and enhanced chemiluminescence (ECL) reagent was purchased from Tanon (Shanghai, China).

### 4.2. Cell Culture, Viability Detection and SiRNA Treatment

HEK 293 cells were cultured with DMEM, 10% FBS, 0.9% l-glutamine, and 1% penicillin/streptomycin in a humidified incubator (Thermo Fisher) at 37 °C, with 5% CO_2_. The cells were exposed to l-proline (1 g/L), AFB1 (100 mg/L), AFM1 (100 mg/L), AFB1 (100 mg/L) + l-proline (1 g/L) and AFM1 (100 mg/L) + l-proline (1 g/L), respectively. After they were co-cultured for 48 h, the cell viability was quantified using a CCK-8 kit to select the appropriate doses for further experiments.

In the PRODH SiRNA treatment experiment, the SiRNA sequences were synthesized by genepharma (Shanghai). HEK 293 cells were seeded into six-well plates for 24 h to promise 40% intensity; DNA-liposome complex was prepared as follows: (1) transfection reagent (5 μL/well, Santa Cruz) was diluted in 1 mL fresh medium; (2) PRODH SiRNA segments (5 μg/well) were diluted in 1 mL fresh medium; (3) the complex of (1) and (2) were mixed together and DNA-liposome complex (2 mL per well) was added into the wells and were placed at 37 °C. 24 h later, the medium-DNA-liposome complex was added into the wells to culture for another 24 h [15]. The step of l-proline treatment also started after removing of the medium-DNA-liposome complex, and the treatment time was 48 h. 

### 4.3. Cell Apoptosis Detection

After AFB1/AFM1/AFB1 + l-proline/AFM1 + l-proline treatments, HEK 293 cells were gathered and washed with icy PBS buffer and were suspended in 100 μL binding buffer. The cell samples were incubated with 10 μL FITC-annexin V buffer and 10 μL PI (10 mg/mL), then gently mixed together and incubated for 15 min at 25 °C in the dark. 200 μL binding buffer per tube was added into the cell samples, and the cells were measured with flow cytometry (BD, Franklin, NJ, USA). Dead (i.e., not alive) cells are all propidium iodide (PI) positive, irrespective of the cause, apoptosis or necrosis and should appear in the late apoptotic (UR) part, cells just undergoing apoptosis are PI negative and are in the early apoptotic (LR) quadrant.

### 4.4. Animal Model

CD-1 mice were purchased from Beijing Vital River Laboratory Animal Technology Co., Ltd. (Beijing, China), with license number SCXK 2012-0001. Animals were fed in cages at 25 °C with a relative humidity of 55%. The mice were acclimatized for at least seven days before commencement. All procedures for animal experimentation were performed according to Chinese guidelines for animal care, confirming to internationally accepted principles in the care and use of experimental animals (NIH publications No. 8023, in 1978). Animal experiments were approved by the Ethics Committee of Chinese Academy of Agriculture Sciences (Beijing, China), with the permission code of “CAS20181015 (Date: 15/10/2018)”. Thirty CD-1 mice (20 ± 2 g, male) were randomly divided into six groups: a control group (untreated), a 10 g/kg b.w. l-proline group, a 0.5 mg/kg b.w. AFB1 group, a 3.5 mg/kg b.w. AFM1 group, AFB1 (0.5 mg/kg b.w.) + l-proline (10 g/kg b.w.) group, and AFM1 (3.5 mg/kg b.w.) + l-proline (10 g/kg b.w.) group, five mice were in each group. AFB1 and AFM1 were dissolved in DMSO/ddH_2_O (final 1%/99%) [35]. The mice in the treatment groups were gavaged once per day (0.2 mL/mice) for 28 days, then sacrificed on day 29. The kidney tissue was dissected and frozen in liquid nitrogen for subsequent histopathology and analysis for biochemical indicators.

### 4.5. Histopathological Analyses

Kidney tissue was isolated and fixed in 4% paraformaldehyde (Solarbio) for 24 h before paraffin embedding and sectioning using a microtome (Leica, Germany). The sections were stained with HE and the histopathology assessed under a light microscope (Olympus, Japan), with photographs being taken at 200 × magnification. To quantify the degree of kidney tissue damage, the scoring system by Image J software (https://sourceforge.net/projects/ihcprofiler) was employed, then the IHC figures were scanned and read. The tissue damages were divided into four degrees, “0–1” stood for negative without any damage, “1–2” stood for low positive (slight inflammatory infiltration), “2–3” stood for positive (visible infiltration and edema), “3–4” stood for high positive (serious edema and hemorrhage).

### 4.6. Biochemical Analysis

Kidney tissue samples were homogenized and centrifuged to collect the supernatant (10 min at 5,000 rpm and 4 °C), kidney tissue and mouse serum samples were utilized for the detection of biochemical indicators, including creatinine, uric acid (UA), urea (UREA), malondialdehyde (MDA) and total antioxidant capacity (T-AOC), which was performed utilizing ELISA kits of these indicators in mouse kidney tissue and mouse serum (Jiancheng, Nanjing, China).

### 4.7. Western Blotting Analysis

The total protein in cells or kidney tissue was extracted by a protein extraction kit (Solarbio, Beijing, China). After heat treatment (95 °C, 10 min), the protein samples were separated on a 10% lauryl sodium sulfate (SDS)-polyacrylamide gel, the proteins were transferred onto a nitrocellulose membrane with Trans-Blot machines (Bio-Rad), and the membrane was blocked with 2.5% BSA in PBST buffer for 1 h at 25 °C. The membranes were then incubated with primary antibodies (β-actin, PRODH, Bax, Bcl-2, cleaved Caspase-3) at 4 °C overnight, with β-actin being used as an internal reference to confirm equal loading. After three washes with PBST buffer (15 min × 3), the membranes were incubated with secondary antibodies at 37 °C for 1 h and then washed (15 min × 3). Finally, specific protein bands were detected using ECL reagent and analyzed using Image J software (Java 1.8.0_172 version, Rawak Software, Inc., Stuttgart, Germany) [36,37].

### 4.8. Statistical Analysis

All the data are presented as mean ± SD. Data analysis was performed using GraphPad Prism 6.0 software (GraphPad, San Diego, USA). Statistical analysis was conducted using Student’s *t*-test and One-Way Analysis of Variance (ANOVA). *p* < 0.05 was considered to indicate a statistically significant difference between the control and treatment groups.

## Figures and Tables

**Figure 1 toxins-11-00226-f001:**
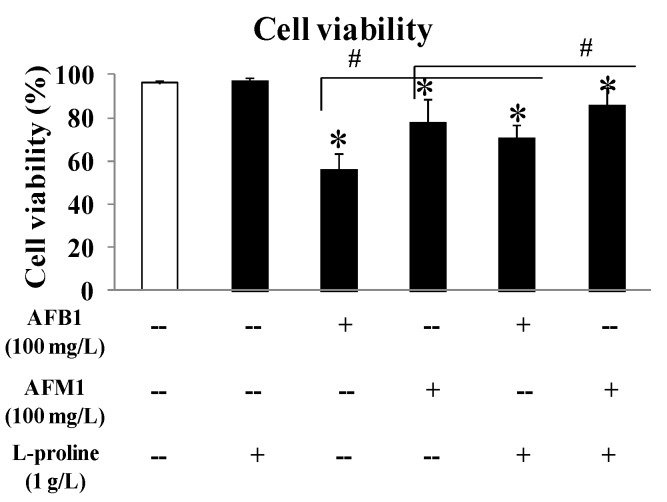
Comparison of human embryonic kidney 293 (HEK 293) cell viability rate and cell death rate affected by aflatoxin B1 (AFB1), aflatoxin M1 (AFM1), AFB1 + l-proline and AFM1 + l-proline. The viability rate and cell death rate were represented as mean ± SD; * *p* < 0.05, compared with control (*n* = 8); ^#^
*p* < 0.05, compared with single aflatoxin group (*n* = 8).

**Figure 2 toxins-11-00226-f002:**
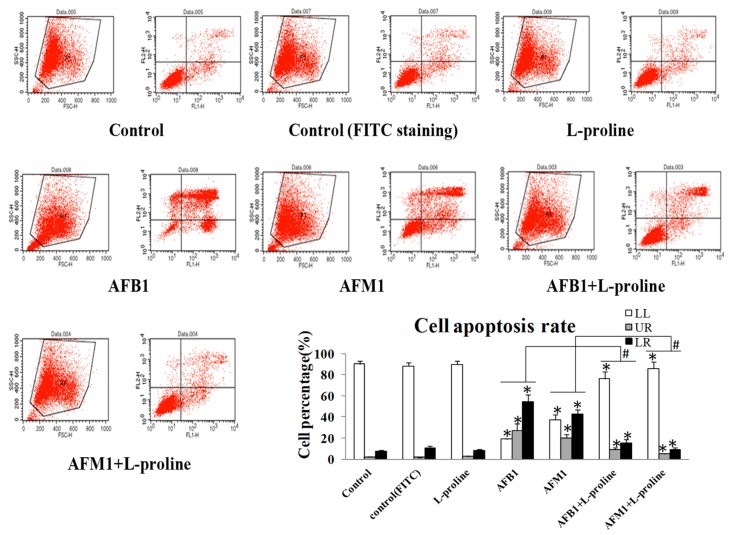
Apoptosis rate of HEK 293 cells affected by AFB1, AFM1, AFB1 + l-proline and AFM1 + l-proline. LL stands for alive cells, LR stands for early apoptosis cells, UR stands for late apoptosis cells and dead cells, respectively. The data was represented as mean ± SD, * *p* < 0.05, compared with control (*n* = 3), ^#^
*p* < 0.05, compared with single aflatoxin group (*n* = 3).

**Figure 3 toxins-11-00226-f003:**
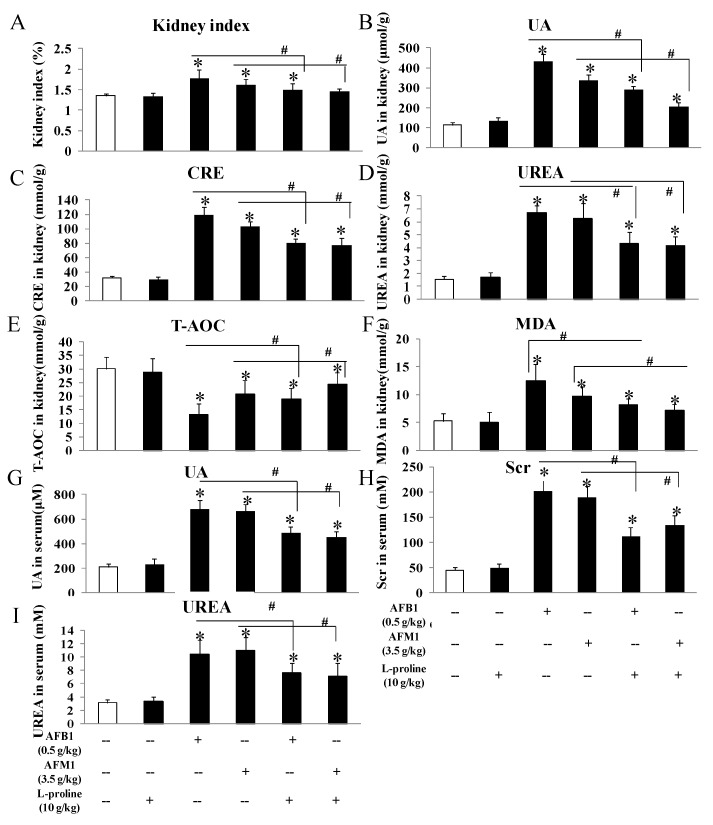
The protective effects of l-proline in kidney damage caused by AFM1 and AFB1: kidney index and biochemical indicators in kidney tissue. (**A**) Kidney index, which was calculated as “(kidney weight/body weight) × 100%”. (**B**–**F**) Content levels of uric acid (UA), creatinine (CRE), urea, malondialdehyde (MDA) and total antioxidant capacity (T-AOC) in kidney tissue. (**G**–**I**) Content levels of UA, CRE and urea in serum. All data were represented as mean ± SD, * *p* < 0.05, compared with control groups; ^#^
*p* < 0.05, compared with AFB1 or AFM1 treatment group (*n* = 5).

**Figure 4 toxins-11-00226-f004:**
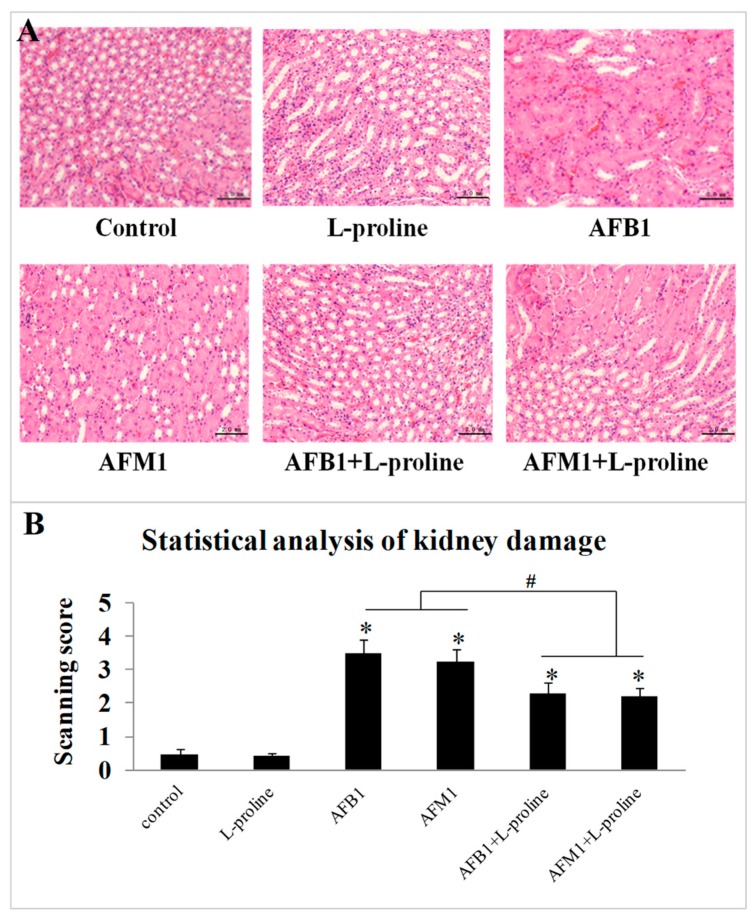
Pathological kidney tissue detection by pathological staining with hematoxylin and eosin. (**A**) Pathological pictures were captured at 200× magnification, in each group, 10 immunohistochemistry (IHC) figures were chosen. (**B**) The pathological pictures in each group were analyzed through scoring system scanning, and the scanning score was represented as mean ± SD, *n* = 10. * *p* < 0.05 was regarded as a significant difference.

**Figure 5 toxins-11-00226-f005:**
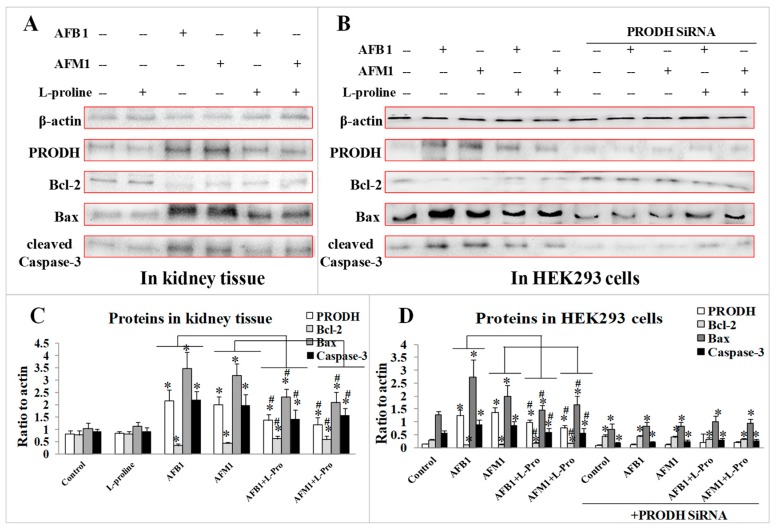
Expression of proline dehydrogenase (PRODH)/Bcl-2/Bax/cleaved Caspase-3—detected by western blotting. (**A**) PRODH/Bcl-2/Bax/cleaved Caspase-3 in kidney tissue. (**B**) PRODH/Bcl-2/Bax/cleaved Caspase-3 in HEK 293 cells treated with PRODH SiRNA. (**C**) Quantification of proteins expression in kidney tissue. (**D**) Quantification of proteins expression in HEK293 cells. All the data was represented as mean ± SD, * *p* < 0.05, compared with control, ^#^
*p* < 0.05, compared with single aflatoxin group, *n* = 3.

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
