# Peer review of "l-Proline Alleviates Kidney Injury Caused by AFB1 and AFM1 through Regulating Excessive Apoptosis of Kidney Cells"

_toxins, 2019, doi:10.3390/toxins11040226_

Round 1

Reviewer 1 Report

Referee evaluation of the manuscript “L-proline alleviates kidney injury caused by AFB1 and AFM1 through regulating excessive apoptosis of kidney cells” (Ref. no.: TOXINS-464667)

Summary:

This manuscript is testing anti-toxic effect of L-proline on kidney and HEK 293 cells, when treated with two aflatoxins.

Hypothesis:

The deleterious effect of two carinogenic mycotoxins is lowered or partially contolled by L-proline.

Novelty:

Indeed the hypothesis is likewise novel. Anyway, mouse and human cells are used as a mixed model. Moreover, the biochemical effect of L-proline are less novel.

Comments:

At many-many cases the text is written in a very confusing way. English correction is essential to understand the Authors intentions.

e.g.:

Introduction

“Referring to the toxic organs”

“multiple of researches”

“Utilizing metabonomics detection of mice kidney”

“metabolic sponsor”

Results

“and results demonstrated that L-proline of 1 g/L seemed no obvious effect on cell viability (97.4%) comparing with the control. At the same concentration (100 mg/L)” - please clarify.

- please do not use axis lablels of 120% for cell viability (Fig. 1)

Fig. 2.: The FCM images are far to small and all details are missing to understand the information content of those. If the bar graph is equivalent with the data, raw FCM dotplots are not needed.*

“and AFB1 and AFM1 treatment caused increasing expressions of CRE, UREA and UA” - these are metabolites and are not expressed per se.

“MDA up-regulated markedly higher” - please clarify.

“and L-proline showed the significant effects” - please clarify.

Discussion

“AFB1 and AFM1 was proved to excrete mainly through the biliary pathway” - grammatically not correct and it is not a pathway. Otherwise, if these AFs are excreted through the bile, why was the kidney chosen?

“MDA is reported to be a peroxide” (27) - how can malondialdehyde be a peroxide? (Additionally, I checked the referenced paper. The statement is not to be found there.)

L165-170 - this description is fully genaral and has no specific meaning.

L171-180 - this is a less relevant section on peroxidation.

My basic problem with the discussion is that it is really mostly focused on the oxidative stress alleviating prporties of L-proline.

Conclusion:

“In summary, we have identified L-proline as the key metabolite of AFB1 and AFM1 in mice kidney tissue” - this is fully erroneous.

“We have also shed light on the activity of the upstream sensor PRODH, which regulates the level of L-proline” - this is the basic function of the enzyme.

Summary: due to the poor writing, not robust basic hypothesis and weak and confuse interpretation I do not recommend this MS for TOXINS as a publication candidate.

*Specific comments on FCM:

“Cells stained annexin V-/PI+ stand for necrotic cells (UL), annexin V+/PI+ cells stand for late apoptotic cells (UR), annexin V+/PI- cells stand for early apoptotic cells (LR). UL - UR - LL - LR quadrants.”

https://www.miltenyibiotec.com/upload/assets/IM0000608.GIF?bust=08bf4b33

Checking the test, 3 populations should be sparated and detected.

The UL population is not valid. Dead, i.e. not alive cells are all PI positive, irrespective of the cause, apoptosis or necrosis and should appear in the UR part, since annexin crosses the damaged membrane and binds to inracellular phospholipids.

Cells just undergoing apoptoisis are PI negative and are in the LR quadrant, since annexin binds to P-serine, appearing in the outer membrane leaflet during the progression of apoptosis. The figure legend is thus simply not enough.

If 4 populations are characteristic for HEK 293 cells, please reference the information, but a more plausible suggestion is debris or non-specific fluorochrom binding.

/* Layout-provided Styles */ div.standard { margin-bottom: 2ex; }

Author Response

1 In “Introduction” part

“Referring to the toxic organs”

“multiple of researches”

“Utilizing metabonomics detection of mice kidney”

“metabolic sponsor”

Answer: We have revised the unclear expression sentences in this part, to make it more comprehensive, including:

1) “Referring to the toxic organs” has been deleted;

2) “multiple of researches” has been revised to “several researches”;

3) “Utilizing metabonomics detection of mice kidney” has been revised to “The mice kidney was detected by metabonomics technology”;

4) “metabolic sponsor” was corrected to be “L-proline was found to be down-regulated significantly is selected as the specific metabolic sponsor in kidney”.

2 In “Results” part

“and results demonstrated that L-proline of 1 g/L seemed no obvious effect on cell viability (97.4%) comparing with the control. At the same concentration (100 mg/L)” - please clarify.

Answer: We have added some information of this experiment, to make it more clear and comprehensive.

3 - please do not use axis lablels of 120% for cell viability (Fig. 1)

Answer: We have corrected the Figure 1, which using the axis labels of 100% as the highest one.

4 Fig. 2.: The FCM images are far to small and all details are missing to understand the information content of those. If the bar graph is equivalent with the data, raw FCM dotplots are not needed.*

Answer: We have adjusted the FCM images, to promise the resolution to be bigger and the images to be more clear.

5 “and AFB1 and AFM1 treatment caused increasing expressions of CRE, UREA and UA” - these are metabolites and are not expressed per se.

Answer: We have revised the sentences here, as “…the levels ofcaused increasing expressions of CRE, UREA and UA in AFB1 and AFM1 treatment groups were higher than the control…”.

6 “MDA up-regulated markedly higher” - please clarify.

Answer: We have clarified and revised the sentence, to make it more comprehensive.

7 “and L-proline showed the significant effects” - please clarify.

Answer: We have corrected the sentence to be “…which indicated thatand L-proline might be helpful showed the significant effects in alleviating oxidative…”.

8 In “Discussion” part

“AFB1 and AFM1 was proved to excrete mainly through the biliary pathway” - grammatically not correct and it is not a pathway. Otherwise, if these AFs are excreted through the bile, why was the kidney chosen?

Answer: We have checked the expression carefully, and revised the sentence to be “…the biliary duct…”. Meanwhile, though AFs are excreted through the bile and the liver, AFs are also be excreted through the kidney, thus, we further emphasized this point in the sentence, to make it more comprehensive.

9 “MDA is reported to be a peroxide” (27) - how can malondialdehyde be a peroxide? (Additionally, I checked the referenced paper. The statement is not to be found there.)

Answer: We have checked the sentence and revised the inaccurate expression, as “…produced from peroxidation of lipids and is proved to be toxic in organ tissue…”. We feel so sorry for the carelessness.

10 L165-170 - this description is fully genaral and has no specific meaning.

Answer: We have checked these sentences and deleted several fully general ones, to improve the expression to be more meaningful.

11 L171-180 - this is a less relevant section on peroxidation.

Answer: We have checked and revised these sentences, to make it more clear and relevant.

12 “In summary, we have identified L-proline as the key metabolite of AFB1 and AFM1 in mice kidney tissue” - this is fully erroneous.

Answer: We have checked the sentence and revised it carefully, as “…we have identified L-proline as the key sensormetabolite of AFB1 and AFM1 in mice kidney tissue in AFs-induced toxicity model…”.

13 “We have also shed light on the activity of the upstream sensor PRODH, which regulates the level of L-proline” - this is the basic function of the enzyme.

Answer: We have revised the sentence, indeed, “…shed light on…” usually reflects the creative point, and we use “proved” here.

14 *Specific comments on FCM: “Cells stained annexin V-/PI+ stand for necrotic cells (UL), annexin V+/PI+ cells stand for late apoptotic cells (UR), annexin V+/PI- cells stand for early apoptotic cells (LR). UL - UR - LL - LR quadrants.”

https://www.miltenyibiotec.com/upload/assets/IM0000608.GIF?bust=08bf4b33

Checking the test, 3 populations should be sparated and detected.

The UL population is not valid. Dead, i.e. not alive cells are all PI positive, irrespective of the cause, apoptosis or necrosis and should appear in the UR part, since annexin crosses the damaged membrane and binds to inracellular phospholipids.

Cells just undergoing apoptoisis are PI negative and are in the LR quadrant, since annexin binds to P-serine, appearing in the outer membrane leaflet during the progression of apoptosis. The figure legend is thus simply not enough.

If 4 populations are characteristic for HEK 293 cells, please reference the information, but a more plausible suggestion is debris or non-specific fluorochrom binding.

/* Layout-provided Styles */ div.standard { margin-bottom: 2ex; }

Answer: Thank you so much for your meaningful suggestion. Previously, we regarded UL cells as dead cells, because I thought some dead cells (broken cells with debris) already lost the integrity of membrane which would be degraded to debris, and caused that annexin-V failed to bind to phosphatidylserine. And in UR quadrant, the membranes of late apoptosis cells and some necrotic cells were also damaged, leading both PI and annexin-V entered into cells which then stained nucleus and phospholipid. In this case, cell membranes lost integrity but still kept those phospholipid compositions, showing a postive staining of annexin-V FITC. So, we didn't ignore UL part in the present analysis. As a reference (Kun Song, et al., 2007. Intraperitoneal photodynamic therapy for an ovarian cancer ascite model in Fischer 344 rat using hematoporphyrin monomethyl ether.) demonstrated, the authors also separated 4 parts of the cells in methods section, indeed, sometimes we were a little confused whether UL area was valid. In order to distinguish dead cells and cells debris from dead cells, we take the percentage of UL cells into account. Here, the proportions of UL cells was a little high in AFs-treated groups, so we counted the number of UL cells as well.

Therefore, in Fig. 2, we have re-analyzed the data and re-drawn the figure, which promise the three populations of the cell samples you mentioned to be separated.  

In conclusion, we really appreciate your suggestions and consideration. Thank you very much!

Reviewer 2 Report

The authors using the experimental animal model and cell cultures showed that L-proline was able to alleviate abnormal expression of biochemical markers and kidney injuries caused by aflatoxins B1 and M1, and they presented that this is as effect of a decrease of oxidative stress and apoptosis.

From my point of view the weakest part of the manuscript are analyses of histopathological changes in kidneys. Authors wrote that L-proline significantly alleviated expression of biochemical markers (yes, this was presented) and kidney pathological damage, however, this damage was not presented.

Histological changes presented in Figure 4 are hardly visible, the description of changes in the text is very short and unclear. The changes should be described in more detail, the authors should use a numerical scale (degrees of changes) allowing for statistical analyses of described differences observed in experimental groups.

In Figure 3 there  is marker –  “kidney index”. What does it mean?

Author Response

1 From my point of view the weakest part of the manuscript are analyses of histopathological changes in kidneys. Authors wrote that L-proline significantly alleviated expression of biochemical markers (yes, this was presented) and kidney pathological damage, however, this damage was not presented.

Answer: We have replaced the pathological pictures, and added a scoring system to quantify the damage degree, which might be helpful to describe the damages caused by AFs and to investigate the protective effect of L-proline.

2 Histological changes presented in Figure 4 are hardly visible, the description of changes in the text is very short and unclear. The changes should be described in more detail, the authors should use a numerical scale (degrees of changes) allowing for statistical analyses of described differences observed in experimental groups.

Answer: We have added some information about the pathological details, in different groups, to make this part to be more clear and comprehensive.

3 In Figure 3 there  is marker –  “kidney index”. What does it mean?

Answer: The kidney index was used to observe the direct damage of kidney organ by AFs, just as “(kidney weight/body weight) × 100%”. Accordingly, we added the description of this indicator in Fig. 3, to make the manuscript more comprehensive.

In conclusion, we really appreciate your suggestions and consideration. Thank you very much!

Reviewer 3 Report

Dear authors,

as I have some relevant questions on the findings I will render my final judgement after your response.

Review on

“L-proline alleviates kidney injury caused by AFB1 and AFM1 throgh regulating excessive apoptosis of kidney cells.”

Aim:

Potential protective effect of oral applied proline on the toxic effects of oral applied aflatoxin B1/M1on kidney.

Approach:

HEK 293 Cells

* Viability (CCK8)

* Necrosis, early and late apoptosis (FACS)

* Protein expression of proline dehydrogenase, Bcl-2, Bax, cleaved casp 3

* PRODH siRNA: proline dehydrogenase, Bcl-2, Bax, cleaved casp 3

Mouse model

* Kidney histology (kidney index)

* creatinine, uric acid, urea, malondialdehyd, total antioxidant capacity in kidney tissue

* Protein expression of proline dehydrogenase, Bcl-2, Bax, cleaved casp 3 in kidney tissue.

Comment

Line 71: Please add the type of viability test to the legend (CCK8)

Line 77: ..and death rate in AFM1+…

Line 8o: What does LL/UL LR+UR mean? Please include in legend

Line 89: “..of CRE, UREA and UA..” => first use of the abbreviation

Line 98 Fig 3: Legend y-axis in C, E,F => mM UREA / Scr ???? How do you calculate a concentration in mmol/L from tissue (fresh weight?).

Why did the authors used kidney tissue for the detection of creatinine, uric acid and urea. These parameter must be detected in blood (serum/plasma) to get a relevant concentration (Remaining blood volume in the kidney after removal?).

Line 109: The authors employed a scoring system to quantify the tissue damage (edema, infiltration etc.). The method is not described and there is no reference literature in “Materials and Methods”.

Line 125 Fig 5:  Did the authors repeat the western blots shown in Fig 5 (n=?). I could not find the quantification of the bands.

Line 206: “..with ninety-five..

Line 224: “..experiment, the SiRNA..”

Author Response

1 Line 71: Please add the type of viability test to the legend (CCK8)

Answer: We have added the type of cell viability test to the legend in “2.1. L-proline alleviates the decreased viability of HEK 293 cells caused by AFB1 and AFM1” section, to make the expression to be more clear.

2 Line 77: ..and death rate in AFM1+…

Answer: We have added the description of cell death rate in these groups, which makes the manuscript more comprehensive.

3 Line 80: What does LL/UL LR+UR mean? Please include in legend

Answer: We have added the explanation of LL/UL/LR/UR/LR+UR in text and legend.

4 Line 89: “..of CRE, UREA and UA..” => first use of the abbreviation

Answer: We have added the full names of these biochemical indicators in this part.

5 Line 98 Fig 3: Legend y-axis in C, E,F => mM UREA / Scr ???? How do you calculate a concentration in mmol/L from tissue (fresh weight?).

Answer: We have revised the unit of legend Y-axis in C, E, F to “mmol/g” in Fig. 3, we really feel sorry for the carelessness.

6 Why did the authors used kidney tissue for the detection of creatinine, uric acid and urea. These parameter must be detected in blood (serum/plasma) to get a relevant concentration (Remaining blood volume in the kidney after removal?).

Answer: We have measured these biochemical indicators including creatinine, uric acid and urea in previous study, and added the results of the three indicators in Fig. 3. Accordingly, we added the result description in “Results” part and “Discussion” part.

7 Line 109: The authors employed a scoring system to quantify the tissue damage (edema, infiltration etc.). The method is not described and there is no reference literature in “Materials and Methods”.

Answer: We have added the description of the scoring system to quantify the degree of kidney tissue damage in “Materials and Methods” part.

8 Line 125 Fig 5:  Did the authors repeat the western blots shown in Fig 5 (n=?). I could not find the quantification of the bands.

Answer: We have repeated the western blots for three times (n=3), and added the quantification results of the bands in Fig 5, which makes the expression more clear.

9 Line 206: “..with ninety-five..

Answer: We have revised “…with Ninety-five…” to  “..with ninety-five…”.

10 Line 224: “..experiment, the SiRNA..”

Answer: We have added the blank in the two words here. We feel so sorry for our carelessness.

In conclusion, we really appreciate your suggestions and consideration. Thank you very much!

Round 2

Reviewer 1 Report

Reading the ms carefully I feel that most of my critics were assessed and corrected. I still feel the need for English correction, but in a scientfic niche the ms may be acceptable. 

Author Response

Reading the ms carefully I feel that most of my critics were assessed and corrected. I still feel the need for English correction, but in a scientfic niche the ms may be acceptable. 

Answer: We have checked the whole manuscript again, and corrected several English grammar errors, improper words and sentences, to make the text more clear and comprehensive.

In conclusion, we really appreciate your suggestions and consideration. Thank you very much for your help and considerations!

Reviewer 2 Report

Now, histological analyses was performed using the software Image J, and the score of histological changes are presented. However, authors should describe how many IHC figures (what does it mean figures, photos of IHC?) were analyzed from each IHC slides from one mouse. As there are 5 mouse per group, the mean score with SD should be presented as the main result per group, and statistical analyses should be done between analyzed group.

In Material and Method section. 4.5 the title “Histopathological test” should be changed into “Histopathological analyses”.

It is not necessary to repeat the description of score in result and method section. The description in Method section is enough.

Author Response

Reviewer 2

The authors using the experimental animal model and cell cultures showed that L-proline was able to alleviate abnormal expression of biochemical markers and kidney injuries caused by aflatoxins B1 and M1, and they presented that this is as effect of a decrease of oxidative stress and apoptosis.

1 Now, histological analyses was performed using the software Image J, and the score of histological changes are presented. However, authors should describe how many IHC figures (what does it mean figures, photos of IHC?) were analyzed from each IHC slides from one mouse. As there are 5 mouse per group, the mean score with SD should be presented as the main result per group, and statistical analyses should be done between analyzed group.

Answer: We have checked the IHC photos carefully, and added some information, including the full name of IHC (immunohistochemistry), the number of IHC figures from one mouse, the mean score with SD value in each group, and the statistical analyses and description among these groups.  

2 In Material and Method section. 4.5 the title “Histopathological test” should be changed into “Histopathological analyses”.

Answer: We have revised the title “Histopathological test” to “Histopathological analyses” in Material and Method section.

3 It is not necessary to repeat the description of score in result and method section. The description in Method section is enough.

Answer: Yes, we have deleted the repeated description of scanning score in Results section.

In conclusion, we really appreciate your suggestions and consideration. Thank you very much for your help and considerations!

Reviewer 3 Report

Line 77: The authors stated an „…enhanced apoptosis rate (UR and LR) and necrosis rate (UL), when compared..”.  UL is mentioned in the text and legend of Fig. 2, but data are not given in diagram.

“Why did the authors used kidney tissue for the detection of creatinine, uric acid and urea. These parameter must be detected in blood (serum/plasma) to get a relevant concentration (Remaining blood volume in the kidney after removal?).

Answer: We have measured these biochemical indicators including creatinine, uric acid and urea in PREVIOUS STUDY, and added the results of the three indicators in Fig. 3. Accordingly, we added the result description in “Results” part and “Discussion” part.”

What do the authors mean with previous study? Where the measurements done in the same set of animals? Did the authors publish the data elsewhere?

Line 267: The authors cannot analyze creatinine, uric acid and urea from frozen tissue (Fig. 3, B,C,D) These parameters MUST be analyzed in blood plasma, an analysis in frozen tissue does not make sense. As far as I understand, the data from serum (?, G,H,I) were measured in another experiment? 

 I have really problems to reconstruct the source of data and the corresponding methods. This is essential for the acceptance of the mauscript.

Line 282: Please give source and type of the ELISA kits.

Author Response

1 Line 77: The authors stated an „…enhanced apoptosis rate (UR and LR) and necrosis rate (UL), when compared..”.  UL is mentioned in the text and legend of Fig. 2, but data are not given in diagram.

Answer: We understand your meaning, UL region stands for unvalid deviation and UR region stands for late apoptosis cells and dead cells, therefore, we only analyze and describe the three regions (LL, UR and LR) this time. We have checked the text and diagram, and deleted description about UL region, to make the text to be more clear and comprehensible.

2 “Why did the authors used kidney tissue for the detection of creatinine, uric acid and urea. These parameter must be detected in blood (serum/plasma) to get a relevant concentration (Remaining blood volume in the kidney after removal?).

Answer: We have measured these biochemical indicators including creatinine, uric acid and urea in PREVIOUS STUDY, and added the results of the three indicators in Fig. 3. Accordingly, we added the result description in “Results” part and “Discussion” part.”

What do the authors mean with previous study? Where the measurements done in the same set of animals? Did the authors publish the data elsewhere?

Answer: Firstly, we must apologize for our unclear expression in the answer.

The correct expression should be: “We ever measured these biochemical indicator in mice serum in previous study [15], and found that the levels of creatinine, uric acid and urea in serum were up-regulated with the treatment of AFs, proving that AFs caused kidney injury [15]. Therefore, we performed the detection of the three indicators both in mice serum and in kidney tissue in present study, and added the results in Fig.3. Accordingly, we added the result description in “Results” part and “Discussion” part.”

We promise that the measurement done in the present manuscript is never published elsewhere.

Reference 15:

Li, H.; Xing, L.; Zhang, M.; Wang, J.and Zheng, N. The Toxic Effects of Aflatoxin B1 and Aflatoxin M1 on Kidney through Regulating L-Proline and Downstream Apoptosis. Biomed Res Int 2018, 2018, 9074861. 9074861.Detection results of creatinine, uric acid and urea in serum in previous study:

In attachment (word)

Detection results of creatinine, uric acid and urea in serum in present study:

In attachment (word)

3 Line 267: The authors cannot analyze creatinine, uric acid and urea from frozen tissue (Fig. 3, B,C,D) These parameters MUST be analyzed in blood plasma, an analysis in frozen tissue does not make sense. As far as I understand, the data from serum (?, G,H,I) were measured in another experiment? I have really problems to reconstruct the source of data and the corresponding methods. This is essential for the acceptance of the mauscript.

Answer: We should apologize for the unclear expression, for the detection in mouse serum has been finished in the present study.

In our previous study, creatinine, uric acid and urea in mouse serum were detected to prove the effects of AFB1 and AFM1 in kidney damage. Therefore, in present study, these indicators were detected not only in mouse serum, but also in kidney tissue, further to validate the effects of AFs and the protective roles of L-proline in kidney damage.

This time, we have revised this paragragh in 4.6. Biochemical analysis section, and added the information about biochemical detection in mice serum, to make the expression more clear.

4 Line 282: Please give source and type of the ELISA kits.

Answer: We have added the information about the source and type of the ELISA kits in 4.6. Biochemical analysis section.

In conclusion, we really appreciate your suggestions and consideration. Thank you very much for your help and considerations!

Round 3

Reviewer 2 Report

Now, after corrections, the paper could be published.

Reviewer 3 Report

Accepted in the present form.